# Trace Element Provision in Parenteral Nutrition in Children: One Size Does Not Fit All

**DOI:** 10.3390/nu10111819

**Published:** 2018-11-21

**Authors:** Boutaina Zemrani, Zoe McCallum, Julie E Bines

**Affiliations:** 1Clinical Nutrition Unit, Department of Gastroenterology and Clinical Nutrition, The Royal Children’s Hospital, 3052 Melbourne, Australia; zoe.mccallum@rch.org.au (Z.M.); Julie.bines@rch.org.au (J.E.B.); 2Department of Pediatrics, University of Melbourne, 3010 Melbourne, Australia; 3Murdoch Children’s Research Institute, 3052 Melbourne, Australia

**Keywords:** trace elements, parenteral nutrition, preterm infants, children

## Abstract

Routine administration of trace elements is recognised as a standard of care in children requiring parenteral nutrition. However, there is a lack of global consensus regarding trace elements provision and dosing in pediatric parenteral nutrition. This review provides an overview of available evidence regarding trace elements supply and posology in parenteral nutrition in neonates and children. Trace elements provision in children should be tailored to the weight and clinical condition of the child with emphasis on those at risk of toxicity or deficiency. Based on current evidence, there is a need to review the formulation of commercial solutions that contain multiple-trace elements and to enable individual trace elements additives to be available for specific indications. Literature supports the removal of chromium provision whereas manganese and molybdenum supplementation are debated. Preterm neonates may have higher parenteral requirements in iodine, selenium and copper than previously recommended. There is growing support for the routine provision of iron in long-term parenteral nutrition. Further studies on trace elements contamination of parenteral nutrition solutions are needed for a range of trace elements.

## 1. Introduction

Trace elements (TE) are essential nutrients required in small amounts to support normal physiological processes [1]. They are required for optimal growth, development and health [2]. Trace elements should be provided to all patients requiring long-term parenteral nutrition (PN), although some TE may not be required for short-term PN therapy [3]. 

There is a lack of global consensus regarding TE provision in pediatric PN, with varying recommendations from North America [4,5,6], Europe [3,7] and Australia [8]. There are also discrepancies between the recommendations issued by the clinical nutrition societies and the commercial TE products available for use in PN. Current practice in TE prescription remains regionalised, often individual to the unit and hospital pharmacy the children are treated in. Unlike adults, TE provision in PN for paediatric patients requires age and/or size related dosing, thus making comparison with adult recommendations problematic.

This review discusses TE provision and posology in PN in neonates and children (<18 years of age) based on recommendations of international clinical nutrition societies, from relevant published studies and clinical experience in this complex patient cohort. The aim is to provide guidance to the prescribing clinician on TE requirements in a range of clinical scenarios and age groups, for short-term (<4 weeks) and long-term (>4 weeks) PN therapy. 

## 2. Methods

A literature review was conducted using MEDLINE and PubMed to search for relevant articles published in English from January 1970 until August 2018 using the following keywords: Trace elements, zinc, copper, selenium, iodine, iron, manganese, chromium, molybdenum, fluoride and parenteral nutrition. Additional citations were hand-searched. The following types of articles were included: Guidelines and recommendations from international nutrition societies, reviews, position papers, research articles and case studies.

## 3. Trace Elements

Trace elements requirements in PN can vary depending on age and clinical indication. TE dosage may require alteration in patients with increased gastrointestinal or skin losses or with hepatic or renal dysfunction. Premature newborns have specific needs given their rapid postnatal growth and limited stores, due to the lack of micronutrients accretion in the third trimester of pregnancy [9]. 

While the absorption of TE is tightly regulated in the gastrointestinal tract, parenteral supply of TE bypasses this homeostatic barrier leading to a risk of overload if excessive quantities are provided [3]. When calculating parenteral requirements for TE, additional sources of TE should be taken into consideration, mainly contamination of PN solutions and enteral intake although absorption may be erratic in the presence of short bowel syndrome or gut dysfunction. PN solutions may contain various amounts of TE present inadvertently, such as can occur with chromium, manganese and aluminium [1,4,10]. Aluminium is a non-nutrient contaminant and its potential toxicity remains a concern for children on long-term PN [4,10,11,12]. 

### 3.1. Zinc

There is a broad variation in zinc requirements within the pediatric population according to age and clinical conditions [13]. Preterm infants and children with high ostomy losses, diarrhoea, exudative skin disease, burns or hypercatabolism have high zinc requirements due to high losses or limited stores [1,3,4,13,14]. Other conditions associated with increased intestinal epithelial cell turnover, such as inflammatory bowel disease, also predispose to zinc deficiency. 

As current standard TE solutions contain fixed zinc dose, additional zinc should be added to PN solutions for these children. It has been suggested that gastrointestinal zinc losses might be as high as 200 μg//kg/day [13]. In our practice, we double the zinc dose in circumstances associated with high losses while monitoring the zinc level. Zinc toxicity is rare and has only been reported in cases of accidental overdose providing large zinc amounts (>50 to 100 mg/day in adults over a prolonged period of time or >1 g of zinc at once) [2,14,15,16]. In children, data is lacking; in one case report, a neonate has inadvertently received 1000-fold the normal dose of zinc in PN resulting in death [17].

The recommended maintenance doses of zinc in PN are consistent between European, Australian and American guidelines with doses ranging from 50 to 500 μg/kg/day according to age, and maximum doses of 5 to 6.5 mg/ day [3,4,6,8].

### 3.2. Copper

Patients with high gastro-intestinal losses, such as jejunostomies, external biliary drainage, exudative burns or continuous renal replacement therapy have high copper losses [2,4,5,18]. Conversely, cholestasis may result in excessive copper accumulation as copper is mainly excreted through bile [1,2,19,20]. It is generally recommended to reduce copper supplementation in PN for cholestatic children [2,13,19], although few authors found no correlation between cholestasis and serum copper levels [21]. Removal of copper from PN in children may lead to copper deficiency [19,22]. As copper is part of a fixed concentration multi-trace element solution, it is difficult to adjust the dose without impacting on other trace elements. There is a need to formulate individual trace element additives to allow changes in dosage more easily. These considerations are particularly important in the setting of long-term PN children prone to intestinal failure-associated liver disease. Dose adjustment is not required in case of short-term PN.

Assessment of adequacy of copper balance can be challenging. In severe copper deficiency, serum copper and ceruleoplasmin levels are low and reflect the copper status of the body [19]. However, copper status can be difficult to assess in marginal deficiency as these parameters can be normal [19]. Although other tests might be helpful in assessing copper status (24-h urine copper, erythrocyte superoxide dismutase), they can present practical challenges and are not widely available [3]. Clinicians should also be aware that copper and ceruleoplasmin levels are increased in the setting of inflammation [19].

Preterm infants have high copper requirements but are also at risk of biliary stasis [9,23]. The European Society of Pediatric Gastroenterology, Hepatology and Nutrition (ESPGHAN) and the European Society of Parenteral and Enteral Nutrition (ESPEN) 2018 guidelines recommend to double the provision of copper in PN to preterm infants (from 20 to 40 μg/kg/day) [3]. While the previous dose is sufficient to prevent acute copper deficiency, it is thought to be lower than the preterm infant’s actual needs to achieve in utero accretion rates (63 μg/kg/day) [9]. However, studies of copper deficiency in premature cholestatic infants primarily report either a copper-free PN use or a dose of 10 μg/kg/day [22,23,24,25,26]. Reducing parenteral copper supplementation in cholestatic preterm infants may increase the risk of copper deficiency [9,27,28]. Copper supplementation of 20 μg/kg/day did not lead to worsening of liver disease in cholestatic preterm infants [9,27]. The increased requirements, limited stores and potential increased gastrointestinal losses of copper must be balanced against the reduced biliary excretion in preterm babies. Furthermore, low concentrations of serum ceruleoplasmin commonly seen in premature infants may be associated with accumulation of copper in hepatocytes without a concurrent increase in serum copper [23]. Therefore, the provision of 20 μg/kg/day for preterm infants with careful monitoring of copper and ceruleoplasmin levels to adjust the dose, particularly in presence of cholestasis, appears a safe approach. Although copper toxicity is rare, chronic excess copper can potentially cause damage in the liver, brain and kidneys [1,13,19]. The toxic dose of parenteral copper in children is not clearly defined in the literature. A study of 8 home PN adults who received 1.4 mg/day of parenteral copper found major elevations of copper levels in autopsy tissues compared to controls, with 2 patients having liver values comparable to Wilson’s disease [29].

### 3.3. Selenium

Pediatric recommendations for selenium provision in PN, except for preterm infants, are consistent across publications at 2–3 μg/kg/day with a maximum of 60 to 100 μg/day [3,4,30]. Critically ill patients or those with severe burns may have higher requirements due to increased oxidative stress and losses through drains, dialysis, or wounds [1,2,30].

According to a Cochrane review, a parenteral supply of 2 μg/kg/day of selenium is inadequate to maintain cord selenium concentrations in preterm infants, whilst doses of 3 μg/kg/day may prevent a decline in cord levels and doses of up to 5–7 μg/kg/day may be required to achieve concentrations close to those found in healthy breast fed infants [3,31,32,33]. It should be stressed that the data in this Cochrane review were dominated by a large trial from New Zealand, a country with low soil selenium concentrations [31,33]. Parenteral supplementation of selenium to preterm infants significantly reduced the risk of 1 or more episodes of sepsis but was not associated with improved survival, reduction in chronic lung disease, or retinopathy of prematurity [31]. The authors concluded that although doses higher than those previously recommended may be beneficial for infants on PN, this may not be readily translated to all populations [31]. ESPGHAN/ESPEN 2018 guidelines [3] support the increase in selenium provision to preterm infants from 2–3 to 7 μg/kg/day based on the New Zealand study in 2000 [33]. Since previous studies in the 1990s [34], the selenium status in New Zealand and some parts of Australia has improved. Data in the 2000s indicated that selenium status indicators of Australian infants and adults are at the lower end of the international range [35], but may be considered marginal in New Zealanders [36]. The clinical consequences and importance of this in relation to selenium provision in PN are unclear. More robust evidence is required to better determine the best posology of selenium in particular in the Australasian area. Although there are no reports of selenium toxicity in children [3], caution is required in case of renal failure as kidneys are the major route of excretion of selenium [13]. 

### 3.4. Iodine

In North America, iodine is not routinely included in PN [1,37,38,39,40], whereas European [3,7] and Australian [2] trace elements commercial solutions usually contain iodine.

A parenteral dose of 1 μg/kg/day of iodine has been suggested for children by most nutrition societies [3,4,41]. While some studies have revealed adequate iodine status under this dose [42], other authors claim it is not sufficient to meet the needs of children on PN [43,44]. There are pediatric reports of iodine deficiency and hypothyroidism described with iodine-free PN use [37,38,39,40,45], highlighting the importance of iodine substitution given its impact on neurodevelopment. Other authors have reported the absence of iodine deficiency despite absence of supplementation [42], likely secondary to adventitious iodine sources. These latter include skin absorption of topical iodinated disinfectants, contamination of PN solutions and absorption of iodine present in the ingested food [3,9,46]. 

In preterm infants, recommendations for parenteral iodine supplementation vary considerably from 1 to 30 μg/kg/day [3,9,13]. The upper value is based on iodine balance studies performed in healthy preterm infants showing that iodine intakes of ≥ 30 μg/kg/day are required to maintain a positive balance [3,9,43]. The initial recommendations of 1 μg/kg/day took into account the potential for significant absorption through the skin of iodine-containing antiseptics [9,46]. However, since chlorhexidine has largely replaced iodine-based antiseptics [9,37,38], concerns about iodine deficiency arose and iodine doses have been revised upwards [38].

The data above highlight the importance of routine screening for iodine deficiency (urine iodine excretion and thyroid function tests [3,38,47]) and adjusting iodine supplementation especially in children receiving exclusive PN, with no enteral intake, to prevent iatrogenic hypothyroidism. 

### 3.5. Iron

Whenever possible, iron supplementation should be given enterally rather than parenterally, if tolerated [3,48]. Iron is generally not part of pediatric trace elements solutions and is usually not included in short-term PN [3]. Patients on long-term PN are at risk of iron deficiency if oral supplementation is not possible [49]. Practices regarding iron provision for long-term PN dependent children are variable across units. Options include either intermittent iron infusions or a daily maintenance dose of iron dextran in PN.

Because of compatibility concerns with PN components, anaphylaxis and iron overload risk, some centres may be reluctant to include iron in PN [3,13,50]. Studies have shown that use of low molecular weight iron dextran preparations and a maintenance dose (as opposed to a higher treatment dose) considerably reduces the anaphylaxis risk and is safe for administration [13,50]. It is generally stated that iron dextran is incompatible with lipids or all-in-one admixtures causing coalescence of lipid droplets [3,51]. Therefore, iron dextran is generally given on non-lipid day(s). However, reports of patients receiving low maintenance levels of iron dextran in all-in-one admixtures (including lipid emulsions) without any compatibility or stability issues have been brought to our attention. In addition, one study has shown the absence of physical incompatibility when iron dextran is added to total nutrient admixtures at a specific concentration [52]. This area warrants further research for a safe and simplified administration.

The risk of infection associated with parenteral iron supplementation is not well defined in the literature [13,48,50,53]. In children with active infection, careful consideration of risk versus benefit of intravenous iron is required. 

Parenteral doses of 50–100μg/kg/day have been suggested for infants and children with supplementation for preterm infants as high as 200–250 μg/kg/day [3,13,54]. Maximum parenteral dose suggested in stable patients is 1–1.1mg/day [8,13], however the 2018 ESPGHAN/ESPEN guidelines recommend up to 5 mg/day. [3] Regular monitoring of iron studies is crucial to detect both iron deficiency and overload. Iron overload has been reported in a cohort of home PN children receiving a dose of 100 μg/kg/day for a prolonged period with high ferritin levels and iron deposition found in liver biopsies in half of the children [55]. Iron provision provided with enteral nutrition should be factored into the calculation of iron requirements.

With the advent of relatively new intravenous iron formulations, such as iron carboxymaltose, intermittent iron infusions are associated with less adverse events, with a favourable safety profile and ease of administration (infusion over 15 min) [13]. Although there are no published studies on the use of iron carboxymaltose in home PN patients, it is regularly used in our home PN cohort with no reported side effects. However, whilst this approach treats iron deficiency once detected, it does not prevent the development of iron deficiency with concerns regarding neurocognitive outcomes. 

### 3.6. Manganese

As manganese is found widely in nature and is a contaminant in PN solutions, debate continues as to whether it should be supplemented in PN [13,56,57]. While deficiency has virtually not been reported in long-term PN, toxicity is more prevalent in particular neurotoxicity and probably liver dysfunction [3,9,56,57,58,59]. A review of 126 pediatric patients receiving variables doses of manganese in PN (>1 μg/kg/day) showed hypermanganesaemia in two thirds of the patients [60]. A similar observation was made amongst term and preterm infants, with brain magnetic resonance showing increased deposition of manganese in the basal ganglia [61,62]. No data are available regarding the long-term neurological effects of manganese exposure during early life.

Given the evidence of manganese toxicity and the scarcity of deficiencies, manganese dose has been significantly reduced compared to previous doses [21,59,63]. Adult AuSPEN Guidelines have decreased manganese provision from 275 μg/day to 55 μg/day [2]. Most authors and nutrition societies agree that pediatric parenteral manganese dose should be at most 1 μg/kg/day with a maximum of 55 μg/day [3,4,6,57,59]. However, other authors state that the contamination in manganese of PN solutions is sufficient to meet the daily requirements of parenterally fed children without need for additional supplementation [9,13,56,57]. Manganese contamination of PN solutions can reach 5 to 310 μg/L [57]. Further studies are required to determine the safe margin of manganese provision. Contamination levels of manganese should be documented on PN solutions.

As the biliary tract is the major route for excretion of manganese, cholestatic patients are at the highest risk of accumulation and toxicity [1]. Most authors agree to remove manganese supplementation in cholestatic patients [3,4,9]. However, the use of TE preparations with fixed ratios limits the flexibility to allow dose adjustment of manganese without altering other TE. Given the frequency of liver disease in children on long-term PN, solutions with no manganese are required. Regular monitoring of whole blood manganese level is recommended in long-term PN with magnetic resonance imaging in cases of suspected toxicity.

### 3.7. Chromium

Chromium deficiency has not been reported in children but has been described in adults on long-term PN [64,65,66,67]. In the pediatric population, there are more concerns regarding chromium toxicity including reduced glomerular filtration rate (GFR) and renal tubular damage [9,68]. Some authors claim that children can receive ample chromium from the contaminants present in PN solutions that satisfy their requirements without the need for additional supplementation [3,68].

Preterm infants are at greater risk of toxicity [69]. In a randomised controlled study, preterm infants receiving 0.2 μg/kg/day of chromium had significantly higher creatinine values within 3 weeks compared with those without chromium supplementation [69]. In another study, serum and urine chromium concentrations were found to be abnormally high in infants and children receiving 0.2 μg/kg/day of chromium supplementation in PN [70]. Another cohort of children receiving an average of 0.15 μg/kg/day of chromium in PN had serum chromium concentrations 20 times higher than those of non-PN controls, with lower than normal GFR [71]. GFR was significantly inversely correlated with serum chromium concentration and daily chromium intake, although a direct causal effect could not be made [71]. Post-mortem data in home PN adults showed that chromium levels were 10- to 100-fold higher than normal concentrations in all tissues studied compared with controls [29].

ESPGHAN/ESPEN [3] don’t recommend chromium addition in PN whereas the American Society of Parenteral and Enteral Nutrition (ASPEN) [4,72] recommends a reduced supplementation compared to previous recommendations and removal of chromium in patients at risk of toxicity. Chromium supplementation should be avoided in patients with renal failure [1,13]. Chromium contamination of PN solutions has been reported to vary greatly [1,68]. Based on this, it is prudent to have data on chromium contamination of locally sourced PN solutions.

There is a lack reliable of serum chromium monitoring that accurately reflects body stores [13]. Resolution of insulin resistance or glucose intolerance after chromium supplementation remains the best measure to confirm chromium deficiency [2,13,68].

### 3.8. Molybdenum

Provision of molybdenum in PN is controversial. Molybdenum is thought to be a contaminant of PN solutions [2]. There are no reports of molybdenum deficiency in children, although it has been described in one adult receiving long term PN without supplementation who presented with cardiac and neurological symptoms [73]. Toxicity data are not available in humans [74].

ESPGHAN/ESPEN recommend the provision of 1 μg/kg/day of molybdenum in preterm infants and 0.25 μg/kg/day in children receiving long term PN [3,4]. Molybdenum supplementation is not provided by ASPEN [72] or AuSPEN [8]. In our practice, clinical deficiencies in molybdenum have not been encountered in the last 20 years despite absence of supplementation. There are no reliable biochemical markers of molybdenum status. Thus, clinicians should be aware of clinical signs of deficiency.

Available data on molybdenum contamination in PN solutions published many years ago are difficult to apply today due to change in packaging and compounding practices [2,75]. Studies are needed to identify whether the contamination levels are sufficient to meet the needs of children on PN.

### 3.9. Fluoride

Information on fluoride provision in PN in children is scarce. Fluoride is not considered as an essential element, although it may contribute to bone strength and prevention of dental caries [41,76,77,78]. Fluoride deficiency has not been described in children on PN and monitoring of fluoride status is not routinely performed [76,77,78]. However, there is a potential for fluoride toxicity in PN with risk of dental fluorosis and impaired bone quality [76,77]. High serum and urine levels may be an indicator of fluoride toxicity [76]. Children on long-term PN may absorb some fluoride from beverages ingested to compensate for stool losses [76,79]. Fluoride contamination of PN has been reported but can be variable [76,77]. Fluoride is present in some TE products used in Australia and Europe. 

ESPGHAN/ESPEN [3,7], ASPEN [4,6,72] and AuSPEN [2,8] guidelines do not recommend fluoride supplementation of PN for children. ASPEN guidelines [72] note that fluoride supplementation could be beneficial, but that more research is needed to inform a recommendation. Greene et al. [41] suggested that 500 mcg/day of fluoride could be considered in infants requiring long-term PN without significant enteral feeding. Whether fluoride might be useful in reducing the osteopenia that is associated with long-term PN remains speculative [76,78].

## 4. Discussion

Despite consensus that trace elements play an important role in a range of critical metabolic processes, recommendations to guide their supplementation in PN diverge. This is particularly problematic for preterm infants who have limited body stores and whose hepatic synthetic pathways and kidneys are immature. Patients at risk of TE deficiency and toxicity require specific consideration with respect to the dose, formulation and route of administration of TE. As the spectrum of paediatric patients may range from 0.5 kg to 100 kg, it is not surprising that a single TE solution may “not fit all” indications and patients. 

Based on current evidence, TE solutions require revision. For patients requiring short-term PN (<4 weeks) and for stable patients, administration of a multi-TE solution titrated to weight would be easy, safe and appropriate to implement. However, the content of multi-TE solutions available on the market requires adjustment of doses of manganese, copper, zinc, selenium, and removal of chromium and fluoride to reflect available knowledge. Routine addition or monitoring of other trace elements including boron and silicon is not currently recommended [3,4,76]. Preterm infants would benefit from a neonatal TE solution that reflects the specific requirements of this age and minimises the risk of toxicity. There is no solution that is widely available to meet this need. For patients with diseases associated with high risk of TE deficiency or toxicity, mainly abnormal gastrointestinal or skin losses and hepatic or renal dysfunction, individual TE preparations should be available to titrate to the patient needs. 

Although there is alignment of recommendations from the international clinical nutrition societies for some TE, differences remain (Table 1). The composition of pediatric parenteral multi-TE solutions available in Australia and New Zealand is provided in Table 2. Table 3 summarises the suggested changes in pediatric parenteral TE dosing for the development of the Australasian guidelines update.

Trace element supplementation in PN should be guided by a careful monitoring and clinical judgment [3,80]. There are challenges in monitoring the adequacy of TE supplementation. Serum biochemical tests do not accurately reflect body status of TE [9,13]. Serum taken during PN infusion may be contaminated by the PN content, however the time between cessation of PN before blood is taken is not well defined. In addition, many micronutrients are acute phase reactants [1,2,13,18,68,81], therefore their levels cannot be reliably interpreted in the setting of an active infection or inflammation (Table 4) [1,2,13,18,68,81]. C-reactive protein level taken concurrently can assist with interpretation of these results. Finally, the impact of a recent blood transfusion on TE levels is unclear. Therefore, serum levels should not be used as the sole indicators of TE status: clinical factors (patient characteristics, underlying disease and its treatment) and clinical signs of deficiencies or toxicities should all be a part of the assessment of TE status.

TE content in PN solutions may be influenced by local factors, including individual ingredients used in the PN solution (source and product), manufacturing process and storage conditions. PN solutions should be labelled with maximum possible content of contaminants, in particular for chromium, manganese and aluminium. Regulations regarding maximum allowable TE contamination level in PN for are warranted. As most studies on PN contamination were published over 2 decades ago, research reflecting the current PN practices, packaging and contamination levels is recommended. Other areas requiring further research include iron addition in all-in-one PN admixtures and the development of reliable markers of micronutrient status.

## Figures and Tables

**Table 1 nutrients-10-01819-t001:** Comparative summary of recommendations for trace element supplementation in paediatric parenteral nutrition.

	Source	Preterm Infant <3 kg (μg/kg/day)	Infant 3–10 kg (μg/kg/day)	Child/Adolescent >10 kg (μg/kg/day)	Max Dose/Adult Dose (μg/day)
Zinc	*ESPGHAN/ESPEN 2018* [3]	400–500	100–250	50	5000
	*AuSPEN 2014* [2]	NA	NA	3200–6500 μg/day (>15 years)	6500
	*ASPEN 2012,2015* [4,69]	300	100	100	3000–5000
	*Wong 2012* [13]	400	250	50	2500–5000
	*ESPGHAN/ESPEN 2005* [7]	450–500	100–250	50	5000
	*ASPEN 2004* [6]	400	50–250	50–125	5000
	AuSPEN 1999 [8] †	200–425	100–250	30–200	3200–6500
	*ASCN, 1988* [37]	400	100–250	50	5000
Copper	*ESPGHAN/ESPEN 2018*	40	20	20	500
	*AuSPEN 2014*	NA	NA	300–500 μg/day (>15 years)	500
	*ASPEN 2012,2015*	20	20	20	300–500
	*Wong, 2012*	20	20	20	300–500
	*ESPGHAN/ESPEN 2005*	20	20	20	NG
	*ASPEN 2004*	20	20	5–20	500
	*AuSPEN 1999* †	20	20	20–25	1270
	*ASCN, 1988*	20	20	20	300
Selenium	*ESPGHAN/ESPEN 2018*	7	2-3	2–3	100
	*AuSPEN 2014*	NA	NA	60–100 μg/day (>15 years)	100
	*ASPEN 2012,2015*	2	2	2	60–100
	*Wong, 2012*	5–7	2	2	30–60
	*ESPGHAN/ESPEN 2005*	2–3	NG	NG	NG
	*ASPEN 2004*	1.5–2	2	1–2	40–60
	*AuSPEN 1999* †	1.3–2	2–3	2.4	30–120
	*ASCN, 1988*	2	2	2	30
Iodine	*ESPGHAN/ESPEN 2018*	1–10	1	1	NG
	*AuSPEN 2014*	NA	NA	130 μg/day (>15 years)	130
	*ASPEN 2012,2015*	NG	NG	NG	NG
	*Wong, 2012*	30	0–1	0–1 (NG for >40 kg)	NG
	*ESPGHAN/ESPEN 2005*	NG	1 μg/day	1 μg/day	NG
	*ASPEN 2004*	NG	NG	NG	NG
	*AuSPEN 1999* †	0.5–9	0.9	0.5–1	130
	*ASCN, 1988*	1	1	1	1
Iron	*ESPGHAN/ESPEN 2018*	200–250	50–100	50–100	5000
	*AuSPEN 2014*	NA	NA	1100 μg/day (>15 years)	1100
	*ASPEN 2012,2015*	NG	NG	NG	NG
	*Wong, 2012*	200–400	50–100	50–100	1000
	*ESPGHAN/ESPEN 2005*	200	50–100	50–100	NG
	*ASPEN 2004*	NG	NG	NG	NG
	*AuSPEN 1999* †	NG	NG	NG	1100
	*ASCN, 1988*	200	100	NG	NG
Manganese	*ESPGHAN/ESPEN 2018*	≤1	≤1	≤1	50
	*AuSPEN 2014*	NA	NA	55 μg/day (>15 years)	55
	*ASPEN 2012,2015*	1	1	1	55
	*Wong, 2012*	0	0	0	50–100
	*ESPGHAN/ESPEN 2005*	NA	1	1	50
	*ASPEN 2004*	1	1	1	40–100
	*AuSPEN 1999* †	0.77–1	1	1	275
	*ASCN, 1988*	1	1	1	50
Chromium	*ESPGHAN/ESPEN 2018*	0	0	0	5
	*AuSPEN 2014*	NA	NA	10–15 μg/day (>15 years)	15
	*ASPEN 2012,2015*	0.0006	0.0006–0.012	0.2–0.7	10
	*Wong, 2012*	0.05–0.2	0.2	0.2	5–15
	*ESPGHAN/ESPEN 2005*	0	0	0	5
	*ASPEN 2004*	0.05–0.2	0.2	0.14–0.2	5–15
	*AuSPEN 1999* †	0.05	NG	NG	10–20
	*ASCN, 1988*	0.2	0.2	0.2	5
Molybdenum	*ESPGHAN/ESPEN 2018*	1	0.25	0.25	5
	*AuSPEN 2014*	NA	NA	19 μg/day (>15 years)	19
	*ASPEN 2012,2015*	NG	NG	NG	NG
	*Wong, 2012*	NG	NG	NG	NG
	*ESPGHAN/ESPEN 2005*	1	0.25	0.25	5
	*ASPEN 2004*	NG	NG	NG	NG
	*AuSPEN 1999* †	NG	NG	NG	40
	*ASCN, 1988*	0.25	0.25	0.25	5
Fluoride	*ESPGHAN/ESPEN 2018*	NG	NG	NG	NG
	*AuSPEN 2014*	NA	NA	NG	NG
	*ASPEN 2012,2015*	NG	NG	NG	NG
	*Wong, 2012*	NG	NG	NG	NG
	*ESPGHAN/ESPEN 2005*	NG	NG	NG	NG
	*ASPEN 2004*	NG	NG	NG	NG
	*AuSPEN 1999 †*	NG	NG	NG	NG
	*ASCN, 1988*	NG	500 μg/day	NG	NG

ASCN: American Society for Clinical Nutrition, ASPEN: American Society for Parenteral and Enteral Nutrition, AuSPEN: Australasian Society for Parenteral and Enteral Nutrition, ESPEN: European Society of Clinical nutrition and Metabolism, ESPGHAN: Pediatric Parenteral Nutrition of the European Society of Pediatric Gastroenterology, Hepatology and Nutrition, NA: not applicable (adult paper), NG: not given. † has been converted from μmols/kg/day to μg/kg/day with rounding as appropriate.

**Table 2 nutrients-10-01819-t002:** Composition of commercial pediatric trace element solutions available in Australia and New Zealand.

	AuSPEN (Baxter)	RCH (Baxter)	Startrace (Biomed)	Peditrace (Fresenius Kabi)	Addaven (Fresenius Kabi)
Dose	1 mL/kg Max 10 mL	0.1 mL/kg Max 2.5 mL	1 mL/kg Max 15 mL	1 mL/kg Max 15 mL	Children >12 years 10 mL
Composition per 1 mL (μg)					
Zinc	91	1960	250	250	5000
*Max dose*	910	4900	3750	3750	
Copper	38	190.7	20	20	380
*Max dose*	380	477	300	300	
Selenium	3.1	0 †	2	2	79
*Max dose*	31		30	30	
Iodine	6.4	8.88	1	1	130
*Max dose*	64	22.2	15	15	
Manganese	2.2	9.88	1	1	55
*Max dose*	22	25	15	15	
Chromium	0.25	0	0	0	10
*Max dose*	2.5				
Fluoride	0	0	57	57	950
*Max dose*			855	855	
Iron	0	0	0	0	1100
Molybdenum	0	0	0	0	19

AuSPEN: Australasian Society for Parenteral and Enteral Nutrition, RCH: Royal Children’s Hospital. † added separately (2 μg/kg/day, maximum 60 μg/day).

**Table 3 nutrients-10-01819-t003:** Proposed 2018 AuSPEN Guidelines for pediatric trace element provision in parenteral nutrition.

	Preterm Infant <3 kg (μg/kg/day)	Infant 3–10 kg (μg/kg/day)	Child/Adolescent >10 kg (μg/kg/day)	Maximum Dose (μg/day)	Special Considerations
**Zinc**	400	100–200	50	5000	Additional supplementation required in patients with diarrhea, high output ostomies or fistulae, cutaneous losses or with hypercatabolism
**Copper**	20	20	20	500	Higher requirements if high digestive losses, external biliary drainage, burns, or in case of continuous renal replacement therapy Reduction in provision required in case of cholestasis (preterm infants may be an exception)
**Selenium**	3	2-3	2	60–100	Higher requirements in critical illness, burns, continuous renal replacement therapy. Dose reduction may be required in case of renal failure
**Iodine**	1	1	1	130	Preterm infants may require up to 10 to 30μg/kg/day. The dose should be adjusted according to laboratory results.
**Iron**	200	50–100	50–100	1000	Iron provision is not required in short-term PN. For long-term PN, iron should not be mixed with lipid emulsions until further evidence is available
**Manganese**	≤1	≤1	≤1	50	No or minimal manganese supplementation Withhold in case of cholestasis
**Chromium**	0	0	0	5–10 †	Addition of Chromium in PN is not suggested as contamination of PN is sufficient to meet requirements. In case it’s provided in PN, cease in presence of renal failure
**Molybdenum**	0	0	0	5 †	Contamination studies in PN are warranted to ensure adequate provision
**Fluoride**	0	0	0	0	Benefits of supplementation unclear. Further research is required

AuSPEN: Australasian Society for Parenteral and Enteral Nutrition, PN: parenteral nutrition. † This represents the maximum daily amount to be provided by contamination of PN solutions. Addition of chromium or molybdenum in PN is not suggested.

**Table 4 nutrients-10-01819-t004:** Impact of acute phase response on serum trace elements levels.

Trace Elements	Effect of Acute Phase Response
Copper	Increased
Ferritin	Increased
Iron	Decreased
Zinc	Decreased
Plasma Selenium †	Decreased
Chromium	Decreased
Manganese	No effect
Iodine, Molybdenum	Unknown

† Red cell selenium is not affected by acute phase response.

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
