# Peer review of "Trace Element Provision in Parenteral Nutrition in Children: One Size Does Not Fit All"

_nutrients, 2018, doi:10.3390/nu10111819_

Round 1

Reviewer 1 Report

Assessment of adequacy of levels, or clinical signs of overt deficiencies, as well as lab measurements, which are fairly well addressed in the discussion, could  be expanded with a little more detail; see below. I think this will give a bigger and deeper didactic frame to the study.

Would  suggest to perhaps expand specifically on difficulty of assessing Cu+++ levels in serum and Urine.

Would also suggest to consider mentioning other clinical signs of deficiencies of Iron and Zinc, such as pica and specially phacophagia  (craving and consumption of ice) for Zn deficiency, most frequently seen in patients with conditions where there is a major increase in the turn-over of the intestinal epithelium, secondary to inflammation as in Celiac disease and IBD, both Crohn’s and UC.

Author Response

Point 1:

Assessment of adequacy of levels, or clinical signs of overt deficiencies, as well as lab measurements, which are fairly well addressed in the discussion, could be expanded with a little more detail; see below. I think this will give a bigger and deeper didactic frame to the study.

Would suggest to perhaps expand specifically on difficulty of assessing Cu+++ levels in serum and Urine.

Response 1:

The authors have added a comment about difficulty of assessing copper status. The text now reads, line 91 to 97:

“Assessment of adequacy of copper balance can be challenging. In severe copper deficiency, serum copper and ceruleoplasmin levels are low and reflect the copper status of the body. However, copper status can be difficult to assess in marginal deficiency as these parameters can be normal. Although other tests might be helpful in assessing copper status (24-hour urine copper, erythrocyte superoxide dismutase), they can present practical challenges and are not widely available. Clinicians should also be aware that copper and ceruleoplasmin levels are increased in the setting of inflammation. “

Point 2:

Would also suggest to consider mentioning other clinical signs of deficiencies of Iron and Zinc, such as pica and specially phacophagia (craving and consumption of ice) for Zn deficiency, most frequently seen in patients with conditions where there is a major increase in the turn-over of the intestinal epithelium, secondary to inflammation as in Celiac disease and IBD, both Crohn’s and UC.

Response 2:

(i) We have added other causes of zinc deficiency mentioned in your comment. The text now reads, line 64 to 68:

“There is a broad variation in zinc requirements within the pediatric population according to age and clinical conditions. Preterm infants and children with high ostomy losses, diarrhoea, exudative skin disease, burns or hypercatabolism have high zinc requirements due to high losses or limited stores. Other conditions associated with increased intestinal epithelial cell turnover, such as inflammatory bowel disease, also predispose to zinc deficiency.”

(ii) We agree that children with eating disorders, such as pica and pagophagia, might develop micronutrient deficiencies. As these children are usually supplemented via the enteral route, we have not included them in this review focused on trace elements provision in parenteral nutrition.

Reviewer 2 Report

The purpose of this study was to review available evidence regarding trace elements supply in PN in pediatric population. Overall, well written and very practical. Some minor changes are suggested.

Issues the author should consider are outlined below:

Methods:

I suggest to specify in detail the time range used to review the database (not only the upper limit: August 2018) and type of articles included (reviews, research articles, trials, recommendations, standards?)

line 46: comma is missing after "manganese"

ZINC

line 68: I suggest to add information on the toxic (over)dose. 

COPPER

line 100: Suggestion as for line 68

Editorial note: it's worth unifying the way of writing element names (from a lowercase letter)

I wonder why fluoride is not included to the review as it is part of some trace elements solutions? Could Authors reply on this?

Author Response

The purpose of this study was to review available evidence regarding trace elements supply in PN in pediatric population. Overall, well written and very practical. Some minor changes are suggested.

Issues the author should consider are outlined below:

Point 1:

Methods: I suggest to specify in detail the time range used to review the database (not only the upper limit: August 2018) and type of articles included (reviews, research articles, trials, recommendations, standards?)

Response 1:

As parenteral nutrition has started in the late 1960s, we have included articles published from January 1970 to August 2018 in our literature review. The type of articles reviewed includes guidelines and recommendations from international nutrition societies, reviews, position papers, research articles, and case studies.

The text now reads, line 44 to 49:

A literature review was conducted using MEDLINE and PubMed to search for relevant articles published in English from January 1970 until August 2018 using the following keywords: trace elements, zinc, copper, selenium, iodine, iron, manganese, chromium, molybdenum, fluoride and parenteral nutrition. Additional citations were hand-searched. The following types of articles were included: guidelines and recommendations from international nutrition societies, reviews, position papers, research articles and case studies.

Point 2:

Line 46: comma is missing after "manganese"

Response 2:

The comma has been added after “manganese”, line 46.

Point 3:

ZINC

Line 68: I suggest to add information on the toxic (over)dose. 

Response 3:

The toxic dose of zinc during parenteral nutrition infusion in children is not well documented in the literature. We have found one case report of acute zinc toxicity in a neonate who accidently received 1000-fold the normal dose resulting in death. The information below has been added to the manuscript, line 72 to 75:

“Zinc toxicity is rare and has only been reported in cases of accidental overdose providing large zinc amounts (> 50 to 100 mg/day in adults over a prolonged period of time or >1 g of zinc at once). In children, data is lacking; in one case report, a neonate has inadvertently received 1000-fold the normal dose of zinc in PN resulting in death.”

Point 4:

COPPER

Line 100: Suggestion as for line 68

Response 4:

The toxic dose of parenteral copper in children is not clearly reported in the literature. Copper levels measured in autopsy tissues of 8 home PN adults who received 1.4 mg/day of parenteral copper found major elevations of copper in tissues compared to controls, with 2 patients having liver values comparable to Wilson’s disease.

The information below has been added to the manuscript, line 114 to 118:

“The toxic dose of parenteral copper in children is not clearly defined in the literature. A study of 8 home PN adults who received 1.4 mg/day of parenteral copper found major elevations of copper levels in autopsy tissues compared to controls, with 2 patients having liver values comparable to Wilson’s disease.”

Response to Editorial note:

Point 1:

It’s worth unifying the way of writing element names (from a lowercase letter)

Response 1:

The writing of trace elements names has been unified to a lowercase letter.

Point 2:

I wonder why fluoride is not included to the review as it is part of some trace elements solutions? Could Authors reply on this?

Response 2:

Thank you for your comment. A paragraph regarding fluoride provision in PN has been added to the text and to tables 1, 3 and 4.

The paragraph now reads, line 266 to 281:

FLUORIDE:

Information on fluoride provision in PN in children is scarce. Fluoride is not considered as an essential element although it may contribute to bone strength and prevention of dental caries. Fluoride deficiency has not been described in children on PN and monitoring of fluoride status is not routinely performed. However, there is a potential for fluoride toxicity in PN with risk of dental fluorosis and impaired bone quality. High serum and urine levels may be an indicator of fluoride toxicity. Children on long-term PN may absorb some fluoride from beverages ingested to compensate for stool losses. Fluoride contamination of PN has been reported but can be variable. Fluoride is present in some TE products used in Australia and in Europe.

ESPGHAN/ESPEN, ASPEN and AuSPEN guidelines do not recommend fluoride supplementation of PN for children. ASPEN guidelines note that fluoride supplementation could be beneficial, but that more research is needed to inform a recommendation. Greene et al. suggested that 500 mcg/day of fluoride could be considered in infants requiring long-term PN without significant enteral feeding. Whether fluoride might be useful in reducing the osteopenia that is associated with long-term PN remains speculative.